# Unanticipated mechanisms of covalent inhibitor and synthetic ligand cobinding to PPARγ

**Jinsai Shang[1,2]\*, Douglas J Kojetin[1,3,4,5,6]\***

[1]Department of Integrative Structural and Computational Biology, Scripps Research and The Herbert Wertheim UF Scripps Institute for Biomedical Innovation & Technology, Jupiter, United States; [2]School of Basic Medical Sciences, Guangzhou Laboratory, Guangzhou Medical University, Guangzhou, China; [3]Department of Biochemistry, Vanderbilt University, Nashville, United States; [4]Center for Structural Biology, Vanderbilt University, Nashville, United States; [5]Vanderbilt Institute of Chemical Biology, Vanderbilt University, Nashville, United States; [6]Center for Applied AI in Protein Dynamics, Vanderbilt University, Nashville, United States

**\*For correspondence:**
shang_jinsai@gzlab.ac.cn (JS);
douglas.kojetin@vanderbilt.edu
(DJK)

**Competing interest:** The authors declare that no competing interests exist.

## eLife Assessment

This **landmark** study elucidates the intricate structural mechanisms by which both covalent and non-covalent synthetic ligands can co-occupy the binding pocket of the nuclear receptor transcription factor PPARγ. Through a **compelling** integration of structural, biochemical, and biophysical evidence, the authors challenge the reliability of two commonly used covalent inhibitors. These findings have far-reaching implications for the broader field of nuclear receptor research. This work will be of high interest to structural biologists and biochemists exploring ligand interactions within the nuclear receptor superfamily.

**Abstract** Peroxisome proliferator-activated receptor gamma (PPARγ) is a nuclear receptor transcription factor that regulates gene expression programs in response to ligand binding. Endogenous and synthetic ligands, including covalent antagonist inhibitors GW9662 and T0070907, are thought to compete for the orthosteric pocket in the ligand-binding domain (LBD). However, we previously showed that synthetic PPARγ ligands can cooperatively cobind with and reposition a bound endogenous orthosteric ligand to an alternate site, synergistically regulating PPARγ structure and function (Shang et al., 2018). Here, we reveal the structural mechanism of cobinding between a synthetic covalent antagonist inhibitor with other synthetic ligands. Biochemical and NMR data show that covalent inhibitors weaken—but do not prevent—the binding of other ligands via an allosteric mechanism, rather than direct ligand clashing, by shifting the LBD ensemble toward a transcriptionally repressive conformation, which structurally clashes with orthosteric ligand binding. Crystal structures reveal different cobinding mechanisms including alternate site binding to unexpectedly adopting an orthosteric binding mode by altering the covalent inhibitor binding pose. Our findings highlight the significant flexibility of the PPARγ orthosteric pocket, its ability to accommodate multiple ligands, and demonstrate that GW9662 and T0070907 should not be used as chemical tools to inhibit ligand binding to PPARγ.

## Introduction

Peroxisome proliferator-activated receptor γ (PPARγ) is a ligand-regulated nuclear receptor transcription factor that regulates gene expression programs controlling adipogenesis and insulin sensitization. Endogenous PPARγ ligands, which include lipids and fatty acids, bind to an orthosteric pocket within the PPARγ ligand-binding domain (LBD) and function as PPARγ agonists that activate gene programs (*Itoh et al., 2008*; *Kliewer et al., 1997*; *Kliewer et al., 1995*; *Li et al., 2008*; *Malapaka et al., 2012*; *Schopfer et al., 2005*; *Waku et al., 2009*). Synthetic small molecule PPARγ ligands, which include FDA-approved antidiabetic drugs, also bind to the same orthosteric pocket, most of which function as agonists that activate PPARγ-mediated transcription. Endogenous and synthetic ligands were originally thought to compete for binding to the PPARγ orthosteric pocket. However, we previously showed that endogenous and synthetic ligands can cobind to PPARγ, likely due to the large size and flexibility of the orthosteric pocket, and synergistically influence PPARγ structure and function (*Shang et al., 2018*).

The structural mechanism of agonist-induced activation of PPARγ transcription has been revealed by structural biology studies including NMR spectroscopy and crystal structures. In the absence of ligand, the apo (ligand-free) PPARγ LBD dynamically exchanges between two or more structural conformations (*Johnson et al., 2000*). NMR data show that a critical structural element in the LBD called activation function-2 (AF-2) helix (helix 12), which is part of the AF-2 coactivator binding surface, exchanges between active and repressive conformations that can be stabilized upon binding ligand (*Shang et al., 2020*). Agonist binding to the orthosteric pocket stabilizes a solvent exposed helix 12 conformation that enables high-affinity binding of coactivators and increases transcription (*Hughes et al., 2012*; *Shang et al., 2019*). Transcriptionally neutral antagonists and repressive inverse agonists developed from orthosteric agonist ligand scaffolds bind via a similar mechanism of agonists but

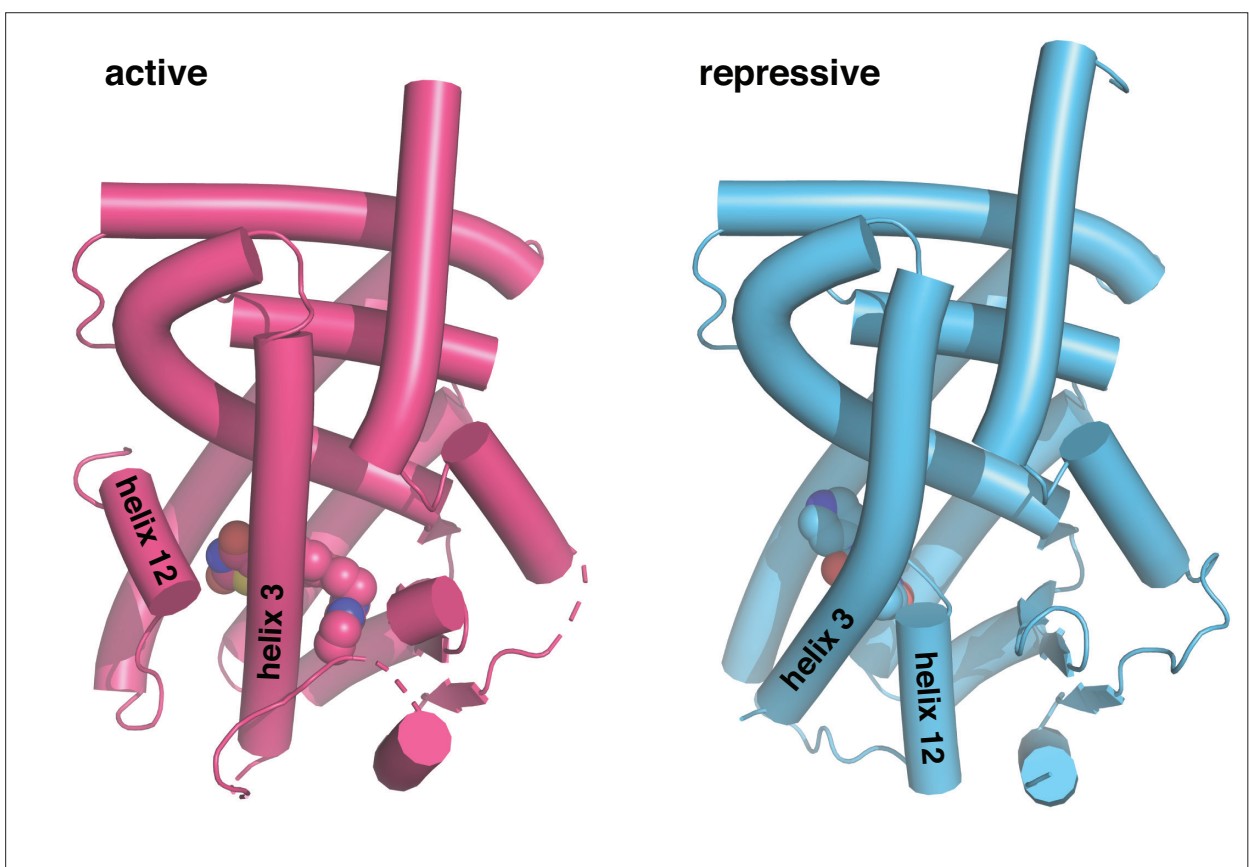

**Figure 1.** Crystal structures of PPARγ LBD in the transcriptionally active and repressive conformations. The active LBD (PDB 6ONJ) is stabilized by agonist (rosiglitazone) and coactivator peptide (TRAP220/MED1), whereas the repressive LBD (PDB 6ONI) is stabilized by covalent inverse agonist (T0070907) and corepressor peptide (NCoR1).

stabilize non-active helix 12 conformations (*Frkic et al., 2023*; *Frkic et al., 2018*; *Marciano et al., 2015*; *Zheng et al., 2018*).

GW9662 and T0070907 are covalent ligands originally described as antagonists as they bind via a nucleophilic substitution mechanism to Cys285 located within the orthosteric pocket and were shown to inhibit binding of select reference PPARγ agonists (*Lee et al., 2002*; *Leesnitzer et al., 2002*). These ligands have been used extensively by the field as covalent antagonist inhibitors to block other synthetic ligands from binding PPARγ to test for synthetic ligand specificity in functional experiments. However, we previously showed GW9662 and T0070907 do not block all ligands from binding to PPARγ (*Brust et al., 2017*; *Hughes et al., 2016*; *Hughes et al., 2014*; *Shang et al., 2018*) — and moreover, they have distinct pharmacological PPARγ functions as a transcriptionally neutral antagonist (GW9662) and repressive inverse agonist (T0070907) (*Brust et al., 2018*). Although these covalent ligands have similar pharmacological functions to orthosteric non-covalent antagonists and inverse agonists, GW9662 and T0070907 function through a different structural mechanism: they slow the rate of exchange between transcriptionally active and repressive conformations natively populated in the apo-LBD, with T0070907 having a more pronounced effect than GW9662 (*MacTavish et al., 2024*; *Shang et al., 2020*). In the transcriptionally repressive conformation stabilized by covalent inverse agonists, helix 12 adopts a solvent-occluded conformation within the orthosteric pocket that overlaps with orthosteric ligand binding poses (*Figure 1*; *Irwin et al., 2022*; *MacTavish et al., 2024*; *Orsi et al., 2022*; *Shang et al., 2020*). These and other published studies have informed a ligand activation model whereby agonist binding to the orthosteric pocket either displaces helix 12 from a solvent occluded repressive conformation within the orthosteric pocket to a solvent exposed active conformation, or selects for an active helix 12 conformation from the dynamic LBD ensemble (*Shang and Kojetin, 2021*).

What remains unclear is the structural basis of covalent inhibitor and synthetic ligand cobinding. This ligand cobinding mechanism was originally discovered in studies of alternate site ligand binding (*Arifi et al., 2023*; *Brust et al., 2017*; *Hughes et al., 2014*; *Hughes et al., 2016*; *Jang et al., 2017*; *Laghezza et al., 2018*; *Leijten-van de Gevel et al., 2022*). Structural studies have mapped the alternate ligand-binding site (*Hughes et al., 2016*; *Hughes et al., 2014*) when two equivalents of a synthetic ligand bind, one to the orthosteric pocket and another to the entrance of the ortho-steric pocket (*Shang and Kojetin, 2021*). NMR and biochemical data revealed non-covalent synthetic compounds can still bind to the PPARγ LBD in the presence of a covalent orthosteric inhibitor (*Brust et al., 2017*; *Hughes et al., 2016*; *Hughes et al., 2014*). While it is presumed that the non-covalent synthetic compound adopts a non-orthosteric binding mode at an alternate site, crystal structures to verify this cobinding mechanism are still needed. Here, using structural biology studies including NMR and crystallography, we confirm that GW9662 and T0070907 do not prevent other synthetic ligands from binding to PPARγ. Furthermore, we demonstrate that certain synthetic ligands can unexpectedly adopt an orthosteric binding pose when cobound with a covalent antagonist inhibitor.

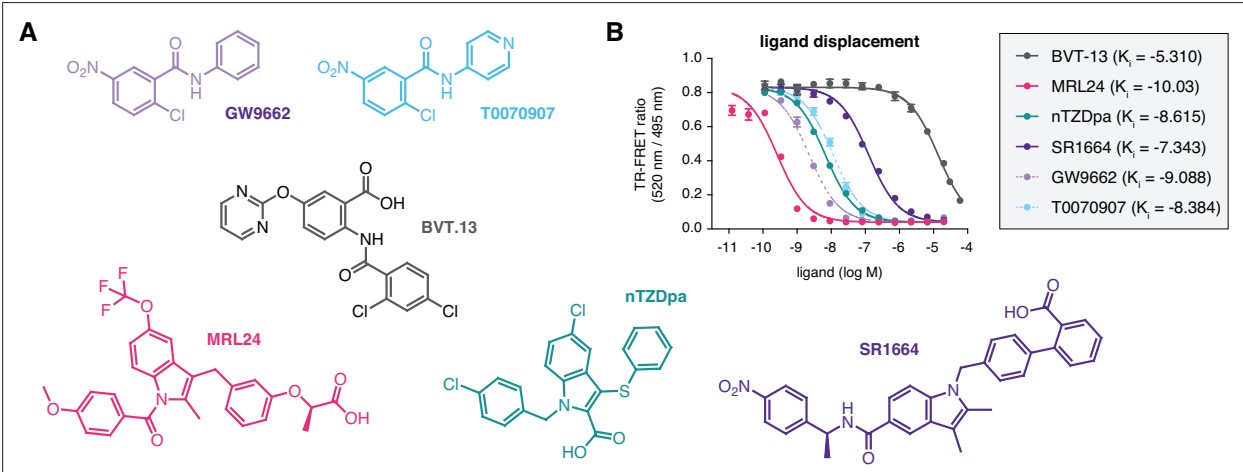

**Figure 2.** Compounds used in the study and relative affinities in a ligand displacement assay. (**A**) Chemical structures of the compounds. (**B**) TR-FRET ligand displacement data for the compounds (n=3; mean ±s.d.).

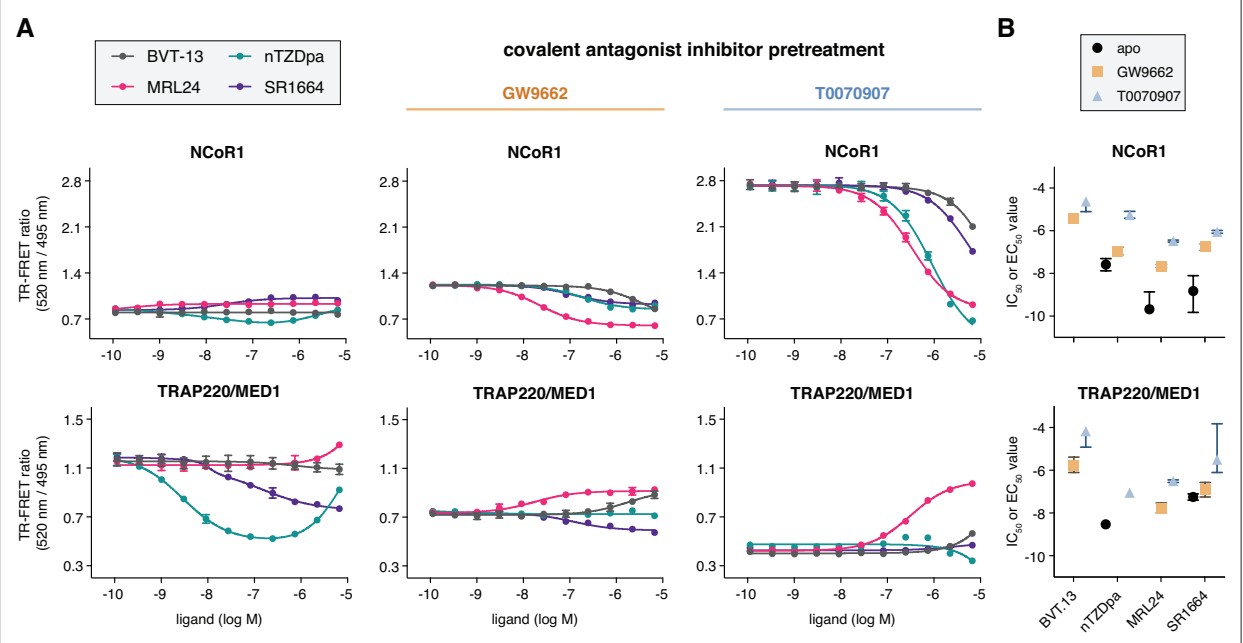

**Figure 3.** Ligand cobinding functional profiling in TR-FRET coregulator interaction assays. (**A**) TR-FRET coregulator interaction assays performed using PPARγ LBD protein with or without preincubation of GW9662 or T0070907 to determine how the non-covalent synthetic ligands influence recruitment of peptides derived from NCoR1 corepressor protein and TRAP220/MED1 coactivator protein fit to a sigmoidal dose response equation or biphasic dose response equation for select cases where a biphasic response is observed (n=3; mean ± s.d.). (**B**) $IC_{50}$ and $EC_{50}$ values extracted from the TR-FRET coregulator interaction data. For curves showing a biphasic response, the higher affinity value is displayed; no value is displayed in cases where the dose response is flat. Error bars when present represent the fitted errors; some fits did not converge to a well-fitted error. See *Figure 3—source data 1*.

The online version of this article includes the following source data and figure supplement(s) for figure 3:

**Source data 1.** TR-FRET coregulator interaction assay data.

**Figure supplement 1.** NMR analysis reveals PPARγ LBD binds more than one equivalent of nTZDpa.

## Results

### Covalent inhibitor and synthetic ligand cobinding influences PPARγ LBD function

We assembled a set of four non-covalent synthetic PPARγ ligands (BVT-13, MRL24, nTZDpa, and SR1664) previously shown to bind the PPARγ LBD via two molar equivalents or bind in the presence of a covalent ligand, GW9662 or T0070907 (*Hughes et al., 2014*; *Figure 2A*). In that previous study, we showed that an analog of MRL24, called MRL20, can activate PPARγ-mediated transcription and increase the expression of PPARγ target genes in differentiated mouse 3T3-L1 preadipocytes when cells were correlated with GW9662 or T0070907 covalent inhibitors. Using a time-resolved fluorescence resonance energy transfer (TR-FRET) biochemical ligand displacement assay, we verified the ligands bind PPARγ LBD with $K_i$ values (*Figure 2B*) consistent with published data (*Acton et al., 2005*; *Berger et al., 2003*; *Choi et al., 2011*; *Ostberg et al., 2004*). These ligands are generally classified as partial agonists that activate PPARγ transcription with limited or weak efficacy, or non-agonists/antagonists that are transcriptionally neutral.

We profiled the non-covalent synthetic ligands using TR-FRET coregulator peptide interaction assays (*Figure 3A*) to determine how the compounds affect interaction between the PPARγ LBD and peptides derived from NCoR1 corepressor and TRAP220/MED1 coactivator proteins, two coregulator proteins that influence PPARγ transcription in cells (*Ge et al., 2002*; *Yu et al., 2005*). Consistent with their partial agonist and/or antagonist profiles, the compounds did not significantly increase interaction with the TRAP220 coactivator peptide. Only two compounds, nTZDpa and SR1664, caused notable changes in the coactivator TR-FRET assay. Of these, nTZDpa showed a biphasic transition that may be due to the binding of more than one nTZDpa molecule, which is also suggested by 2D

[$^1$H,$^{15}$N]-TROSY-HSQC NMR data where chemical shift perturbations (CSPs) are observed going from 1 to 2 equivalents of added ligand.

We next performed TR-FRET coregulator interaction assays using PPARγ LBD protein that was first pretreated with a covalent ligand, GW9662 or T0070907, followed by titration of the non-covalent synthetic ligands. When bound to GW9662 or T0070907, corepressor peptide binding affinity to the LBD is progressively strengthened upon binding to GW9662 and T0070907 while coactivator peptide affinity is weakened according to their neutral and repressive pharmacological activities, respectively (*MacTavish et al., 2024*). As a result of this change in coregulator affinity, the baseline TR-FRET signal progressively increases in the corepressor assay and decreases in the coactivator assay. Titration of the synthetic ligands shows notable changes in the NCoR1 corepressor TR-FRET assay, where the ligands decrease NCoR1 peptide interaction with $IC_{50}$ profiles (*Figure 3B*) with a similar rank-order to ligand $K_i$ values (MRL24 <nTZDpa < SR1664<BVT.13). Relatively minor changes are observed for the ligands in the TRAP220/MED1 coactivator TR-FRET assay except for MRL24, which shows a concentration-dependent increase in coactivator peptide recruitment. Taken together, these data indicate the functional effect of synthetic ligands binding to PPARγ LBD in the presence of a covalent inhibitor is decreased corepressor peptide interaction. Notably, the synthetic ligand $IC_{50}$ values are weakened (right shifted) more by the inverse agonist T0070907 compared to the neutral antagonist GW9662, suggesting that pharmacological repressive ligand efficacy may be involved in the ligand cobinding inhibitory mechanism.

## NMR studies indicate covalent inhibitors allosterically weaken cobinding of non-covalent synthetic ligands by stabilizing a repressive conformation

Two mechanisms may contribute to the covalent inhibitor mechanism of weakening synthetic ligand cobinding. The covalent ligands could structurally overlap or clash with synthetic ligand orthosteric binding modes, leading to alternate site binding with a reduced binding affinity. In this case, the relatively simple phenyl group (GW9662) to pyridyl group (T0070907) change would somehow lead to a more robust clash between the covalent inhibitor and synthetic ligand. Alternatively, the TR-FRET data suggested a different mechanism that involves the repressive efficacy of the covalent ligand. The pharmacological shift from a neutral covalent antagonist (GW9662) to repressive covalent inverse agonist (T0070907) may allosterically shift the dynamic LBD ensemble towards a transcriptionally repressive conformation, where helix 12 adopts a solvent occluded conformation within the orthosteric pocket that structurally clashes with orthosteric binding of a synthetic ligand. In this case, synthetic ligand binding to T0070907-bound PPARγ LBD would significantly influence the NMR-detected repressive LBD conformation where helix 12 is within the orthosteric pocket more than the active LBD conformation where helix 12 is solvent exposed and not occluding the orthosteric pocket.

To structurally assess the non-covalent cobinding mechanism, we performed protein NMR footprinting by comparing 2D [$^1$H,$^{15}$N]-TROSY-HSQC NMR data of $^{15}$N-labeled PPARγ LBD preincubated with a covalent ligand (GW9662 or T0070907) in the absence or presence of a synthetic ligand. Non-covalent ligand cobinding to GW9662-bound LBD shows NMR CSPs for select peaks (*Figure 4A*). In contrast, NMR CSPs are more pronounced for non-covalent ligand binding to T0070907-bound LBD (*Figure 4B*). Focusing on Gly399, a residue near the AF-2 surface that displays two T0070907-bound NMR peaks in slow exchange corresponding to the active or repressive LBD state but only one GW9662-bound active state peak (*Brust et al., 2018*; *MacTavish et al., 2024*; *Shang et al., 2020*), cobinding of the non-covalent synthetic ligand causes the two NMR peaks in slow exchange to converge to one NMR peak. The NMR peaks corresponding to the repressive T0070907-bound conformation disappear, while the remaining peaks have NMR chemical shift values similar to the T0070907-bound active state but shifted along the active-repressive continuum (i.e. diagonal between the active and repressive T0070907-bound NMR peaks) that correlates with function (*MacTavish et al., 2024*).

Among the synthetic ligands we tested, MRL24 cobinding shows an increase in TRAP220 coactivator peptide recruitment to T0070907-bound PPARγ LBD and the largest shift NMR-detected shift of the LBD conformational ensemble towards an active state. In contrast, nTZDpa cobinding shows an antagonist-like profile in the TR-FRET data, decreasing both NCoR1 corepressor and to a lesser degree TRAP220 coactivator interaction, and shifts the NMR-detected shift of the LBD conformational

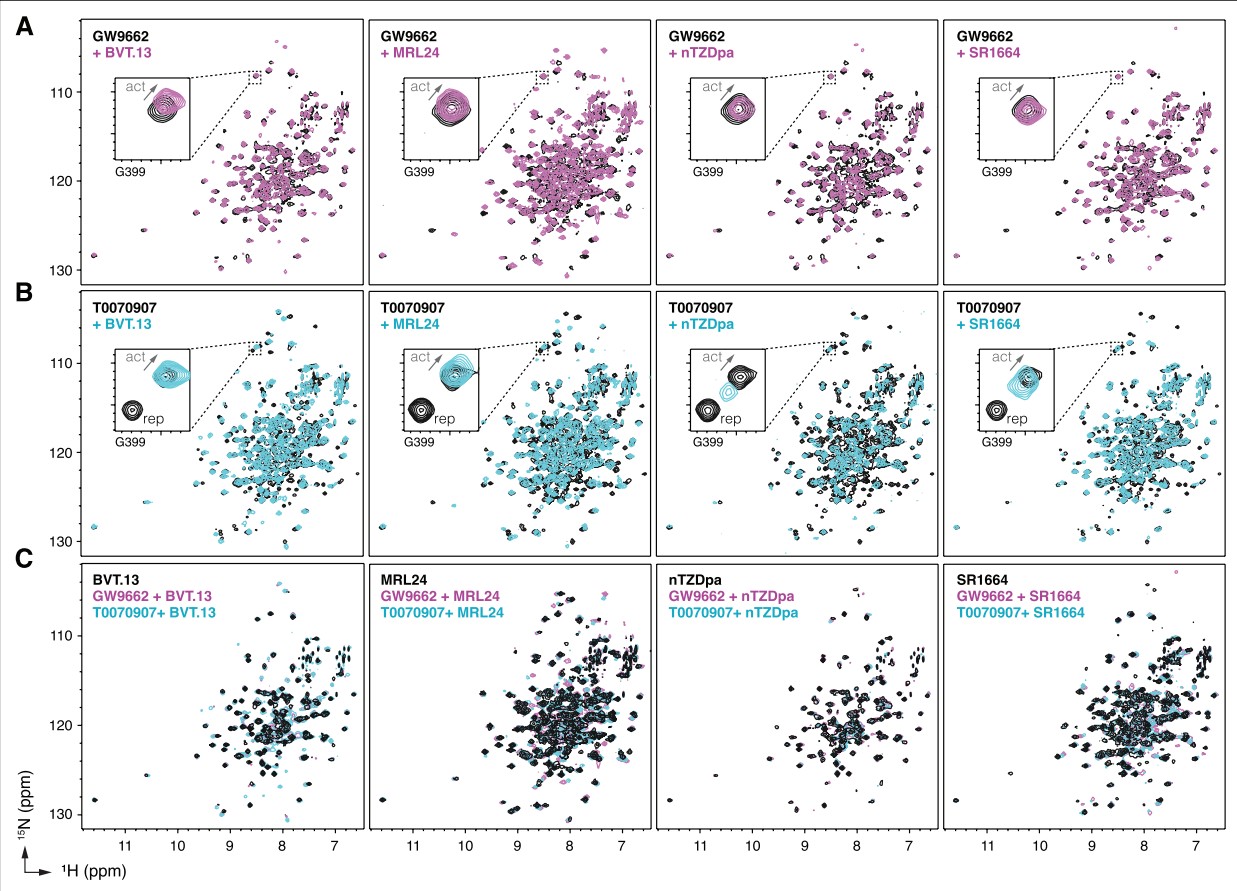

**Figure 4.** NMR implicates covalent inhibitor-induced stabilization of a repressive LBD conformation in the mechanism of weakening non-covalent synthetic ligand cobinding. Overlays of 2D [$^1$H,$^{15}$N]-TROSY-HSQC NMR data of $^{15}$N-labeled PPARγ LBD preincubated with covalent inhibitor, (**A**) GW9662 or (**B**) T0070907, in the absence or presence of the indicated non-covalent synthetic ligands added at 2 molar equivalents. (**C**) Overlays of 2D [$^1$H,$^{15}$N]-TROSY-HSQC NMR data of $^{15}$N-labeled PPARγ LBD in the presence of non-covalent synthetic ligands (singly bound state) compared to the cobound states with a covalent inhibitor.

The online version of this article includes the following figure supplement(s) for figure 4:

**Figure supplement 1.** NMR spectral overlays show chemical shift perturbations (CSPs) between different single ligand-bound PPARγ LBD states.

ensemble towards an intermediate state between the active and repressive T0070907-observed populations.

Taken together, these NMR-detected observations support the mechanism whereby the repressive conformation helix 12 within the orthosteric pocket of T0070907-bound LBD is displaced to a solvent-exposed active conformation. Furthermore, comparison of 2D [$^1$H,$^{15}$N]-TROSY-HSQC NMR data of $^{15}$N-labeled PPARγ LBD bound to the synthetic ligands alone, or cobound with GW9662 or T0070907, show similar spectral profiles with more subtle CSPs (*Figure 4C*) compared to the larger NMR CSPs observed when comparing $^{15}$N-labeled PPARγ LBD bound to each synthetic ligand alone or T0070907-bound LBD relative to GW9662-bound LBD (*Figure 4—figure supplement 1*; *Shang et al., 2020*). This suggests the active conformation of the PPARγ LBD when bound to a synthetic ligand alone vs. cobound to a covalent ligand are similar, which is supported by the TR-FRET data (*Figure 3*) showing that synthetic ligand cobinding to T0070907-bound PPARγ LBD decreases corepressor peptide interaction.

## Crystal structures reveal disparate alternate site ligand binding poses

To visualize the ligand cobinding poses, we first crystallized complexes of PPARγ LBD covalently bound to GW9662 or T0070907, which produced solvent exposed active (chain A) and inactive (chain B) helix 12 conformations similar to apo-PPARγ LBD (*Nolte et al., 1998*), then we added synthetic ligands to

**Table 1.** X-ray crystallography data collection and refinement statistics.

| | PPARγ LBD bound to GW9662 and BVT.13 | PPARγ LBD bound to GW9662 and MRL24 | PPARγ LBD bound to GW9662 and nTZDpa | PPARγ LBD bound to T0070907 and BVT.13 | PPARγ LBD bound to T0070907 and MRL24 | PPARγ LBD bound to T0070907 and nTZDpa | PPARγ LBD bound to T0070907 and SR1664 |
|---|---|---|---|---|---|---|---|
| Data collection[*] | | | | | | | |
| Space group | C 1 2 1 | C 1 2 1 | C 1 2 1 | C 1 2 1 | C 1 2 1 | C 1 2 1 | C 1 2 1 |
| Cell dimensions | | | | | | | |
| $a, b, c$ (Å) | 92.19, 61.99, 118.84 | 91.84, 62.22, 119.24 | 92.75, 62.22, 119.08 | 92.64, 61.78, 119.13 | 92.42, 61.63, 119.55 | 93.02, 62.16, 119.46 | 93.88, 62.68, 121.04 |
| α, β, γ (°) | 90, 102.38, 90 | 90, 102.28, 90 | 90, 102.19, 90 | 90, 102.38, 90 | 90, 102.19, 90 | 90, 102.14, 90 | 90, 102.46, 90 |
| Resolution (Å) | 51.06–2.54 (2.63–2.54) | 49.06–2.48 (2.57–2.48) | 49.16–3.15 (3.26–3.15) | 51.02–2.49 (2.58–2.49) | 58.43–2.56 (2.65–2.56) | 58.39–2.73 (2.83–2.73) | 59.09–3.2 (3.31–3.2) |
| $R_{merge}$ | 0.088 (1.159) | 0.132 (1.791) | 0.043 (0.212) | 0.076 (1.083) | 0.087 (1.367) | 0.108 (1.621) | 0.046 (0.263) |
| $I / \sigma I$ | 12.06 (1.38) | 8.54 (1.04) | 15.02 (3.63) | 14.71 (1.76) | 12.80 (1.42) | 10.28 (1.21) | 10.70 (2.94) |
| Completeness (%) | 98.14 (96.97) | 98.41 (96.93) | 99.59 (100.00) | 99.14 (99.22) | 99.47 (98.55) | 98.79 (98.27) | 99.87 (100.00) |
| Redundancy | 6.6 (6.4) | 6.5 (6.5) | 2.0 (2.0) | 6.6 (6.6) | 6.6 (6.4) | 6.6 (6.7) | 2.0 (2.0) |
| Refinement | | | | | | | |
| Resolution (Å) | 2.54 | 2.48 | 3.15 | 2.49 | 2.56 | 2.73 | 3.2 |
| No. unique reflections | 21761 | 23546 | 11672 | 23269 | 21415 | 17965 | 11555 |
| $R_{work}$ / $R_{free}$ | 25.2/31.6 | 24.2/29.8 | 21.6/29.8 | 23.6/29.5 | 23.7/27.8 | 25.0/30.4 | 20.6/28.3 |
| No. atoms | | | | | | | |
| Protein | 4149 | 4065 | 3921 | 4123 | 4066 | 4096 | 4015 |
| Ligand/ion | 90 | 112 | 74 | 90 | 112 | 46 | 118 |
| Water | 35 | 82 | 2 | 43 | 21 | 18 | 0 |
| B-factors | | | | | | | |
| Protein | 65.80 | 53.94 | 58.48 | 64.00 | 64.71 | 67.70 | 79.58 |
| Ligand/ion | 74.57 | 41.56 | 77.08 | 74.58 | 49.66 | 83.90 | 108.68 |
| Water | 55.23 | 46.34 | 46.73 | 51.89 | 50.58 | 55.19 | n/a |
| R.m.s. deviations | | | | | | | |
| Bond lengths (Å) | 0.010 | 0.011 | 0.013 | 0.010 | 0.010 | 0.012 | 0.011 |
| Bond angles (°) | 1.33 | 1.45 | 1.43 | 1.21 | 1.38 | 1.35 | 1.31 |
| Ramachandran favored (%) | 92.90 | 96.77 | 94.33 | 96.63 | 96.57 | 95.01 | 92.61 |
| Ramachandran outliers (%) | 0.20 | 0.20 | 0.00 | 0.00 | 0.20 | 0.20 | 0.21 |
| PDB accession code | 8ZFN | 8ZFP | 8ZFO | 8ZFQ | 8ZFS | 8ZFR | 8ZFT |

[*]Values in parentheses are for highest-resolution shell.

the crystals using soaking methods. We obtained seven crystal structures in total where each synthetic ligand was cobound to either GW9662 or T0070907, except for SR1664 for which we only obtained a structure cobound to T0070907 (*Table 1*, *Figure 5—figure supplement 1*). In most structures, electron density was observed for non-covalent and covalent ligands in both chains. However, in the nTZDpa structures, the covalent ligand was not observed in chain A; and nTZDpa was observed in

**Table 2.** Structural rmsd comparison of ligand cobound structures to the transcriptionally active PPARγ LBD conformation (PDB 6ONJ).

| PDB ID | rmsd |
|--------|------|
| 8ZFP | 0.98 |
| 8ZFO | 0.77 |
| 8ZFQ | 0.97 |
| 8ZFS | 0.95 |
| 8ZFR | 0.92 |
| 8ZFT | 1.00 |
| 8ZFN | 1.03 |

chains A and B when cobound to GW9662, but only chain B when cobound to T0070907. The structures show high structural similarity to the transcriptionally active LBD conformation with rmsd values ranging from 0.77 to 1.03 Å (*Table 2*).

We compared our cobound crystal structures to published structures of PPARγ LBD bound to non-covalent ligands—BVT.13 (*Bruning et al., 2007*), MRL24 (*Bruning et al., 2007*), nTZDpa (*Bruning et al., 2007*), and SR1664 alone (*Marciano et al., 2015*) or cobound to NCOA1 coactivator peptide (*Bae et al., 2016*) —and alone to covalent ligands—GW9662 (*Shang et al., 2018*) and T0070907 (*Brust et al., 2018*). Overall, the LBD conformations when bound to a single ligand or cobound two ligands are highly similar, with only relatively minor conformational changes in certain residue side chains. Although this is largely influenced by the crystallized forms used for soaking, these findings are also consistent with the aforedescribed 2D NMR data, which indicates a similar LBD conformation for these different liganded states.

Focusing on the non-covalent synthetic ligand cobinding poses (*Figure 5A*), most of the structures surprisingly showed that the ligand adopts a cobound conformation similar to orthosteric binding pose observed in crystals structures of PPARγ LBD bound to the synthetic ligand alone (*Figure 5B*). Slight reorientations of portions of the synthetic ligand occur to accommodate the cobinding mode, as there are clashes between the synthetic orthosteric and covalent orthosteric ligand binding modes (*Figure 5C*). Of note, below we use 'orthosteric binding pose' to refer to the crystallized ligand conformation when PPARγ LBD is bound to a single ligand.

The BVT.13 cobinding pose is similar to its orthosteric binding pose located near the β-sheet. However, the 2,4-dichloro group clashes with the orthosteric GW9662 and T0070907 binding poses, specifically the phenyl and pyridyl groups respectively, resulting in a slight reorientation of the 2,4-dichloro group to accommodate the cobound state. The MRL24 cobinding pose is also similar to its orthosteric ligand binding pose, which was surprising given its larger scaffold size significantly clashes with the orthosteric covalent ligand binding pose. The nTZDpa cobinding pose is also similar to its orthosteric binding pose near the β-sheet. However, the 1-chloro group clashes with the orthosteric covalent ligand binding pose, resulting in a minor reorientation in the cobinding pose. Finally, the SR1664 cobinding pose reveals an alternate site binding mode similar to the crystallized binding pose in the presence of NCOA1 peptide, with a slight reorientation of the benzoic acid group that avoids a clash with the orthosteric T0070907 binding pose. Notably, these alternate site binding modes are distinct from the orthosteric SR1664 binding mode obtained without coregulator peptide, which shows a large steric clash with the orthosteric covalent ligand binding pose.

Focusing on the covalent ligand binding poses, previously determined crystal structures (*Shang et al., 2020*) show that GW9662 and T0070907, when bound alone, are oriented in different directions within the orthosteric pocket in the active state when soaked into apo-PPARγ LBD or repressive state when cobound to NCoR1 corepressor peptide, pointing towards or away from the β-sheet surface, respectively (*Figure 6A*). Cobinding of a non-covalent synthetic ligand has different effects on the covalent ligand binding pose (*Figure 6B*). For BVT.13, the covalent ligand cobinding pose is similar to the orthosteric binding pose. For MRL24, the cobound covalent ligands adopt a similar binding pose that is different from the active and repressive orthosteric binding poses. For nTZDpa, the cobound covalent ligands adopt different orientations, both of which are distinct from the active and repressive covalent ligand only-bound states. For SR1664, the cobinding mode of T0070907 is similar to the orthosteric binding mode. These structures show that the conformation of the covalent ligand can change and adapt to the cobound non-covalent ligand.

Notably, orthosteric ligand $K_i$ values for the non-covalent compounds (*Figure 2B*) correlate with the ability of the non-covalent ligand to push the covalent ligand into a cobinding binding mode that is distinct from the active and repressive covalent ligand binding modes (*Figure 6B*). MRL24 and nTZDa,

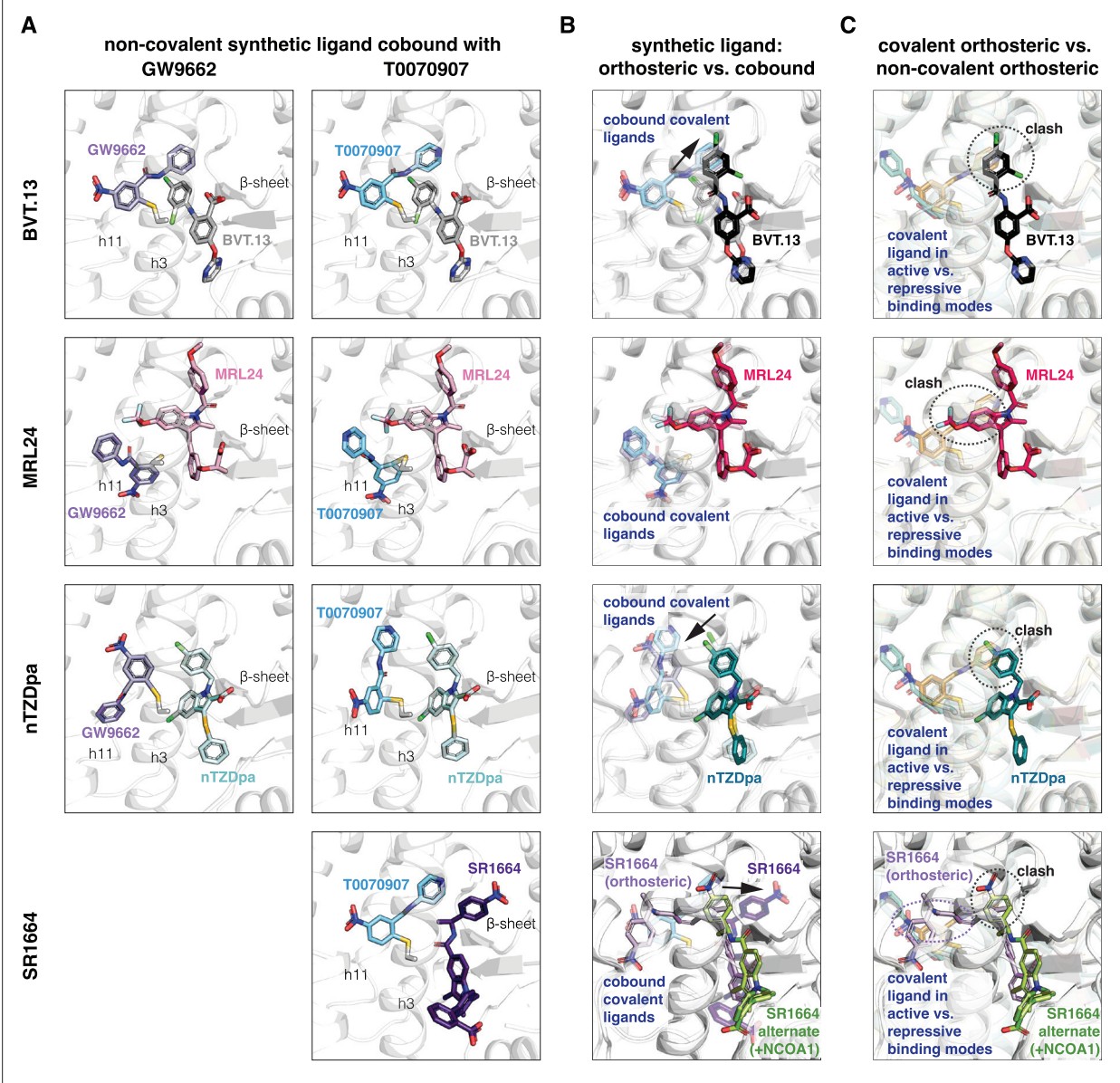

**Figure 5.** Non-covalent synthetic ligands adopt orthosteric binding modes when cobound with a covalent inhibitor. (**A**) Ligand cobinding modes in crystal structures of PPARγ LBD. (**B**) Comparison of the non-covalent synthetic ligand orthosteric binding mode (singly bound) and ligand cobinding mode with a covalent inhibitor (transparent sticks). Differences between these binding modes are indicated with a black arrow. (**C**) Structural clashes observed between the covalent inhibitor orthosteric binding mode (transparent sticks) and the non-covalent synthetic ligand binding mode. PDB codes for crystal structures used in the overlays are listed in the Materials and methods section.

The online version of this article includes the following figure supplement(s) for figure 5:

**Figure supplement 1.** Electron density is shown from composite omit $2F_o$-$F_c$ maps (contoured at 0.8-1σ) of ligands for PPARγ LBD cobound to non-covalent ligands.

which are the most potent non-covalent ligands tested, pushed the covalent ligands into different non-natural conformations, whereas BVT.13 and SR1664 do not. Taken together with structural data showing that the covalent ligands naturally exchange between different active and repressive binding poses (**Brust et al., 2018**; **Shang et al., 2020**), these findings indicate the malleability of the dynamic orthosteric pocket, the natural orthosteric binding mode of the non-covalent ligand, and the relative orthosteric affinity of the non-covalent ligand may determine whether the covalent ligand traps a non-covalent ligand in the entrance to and/or the β-sheet region of the orthosteric pocket.

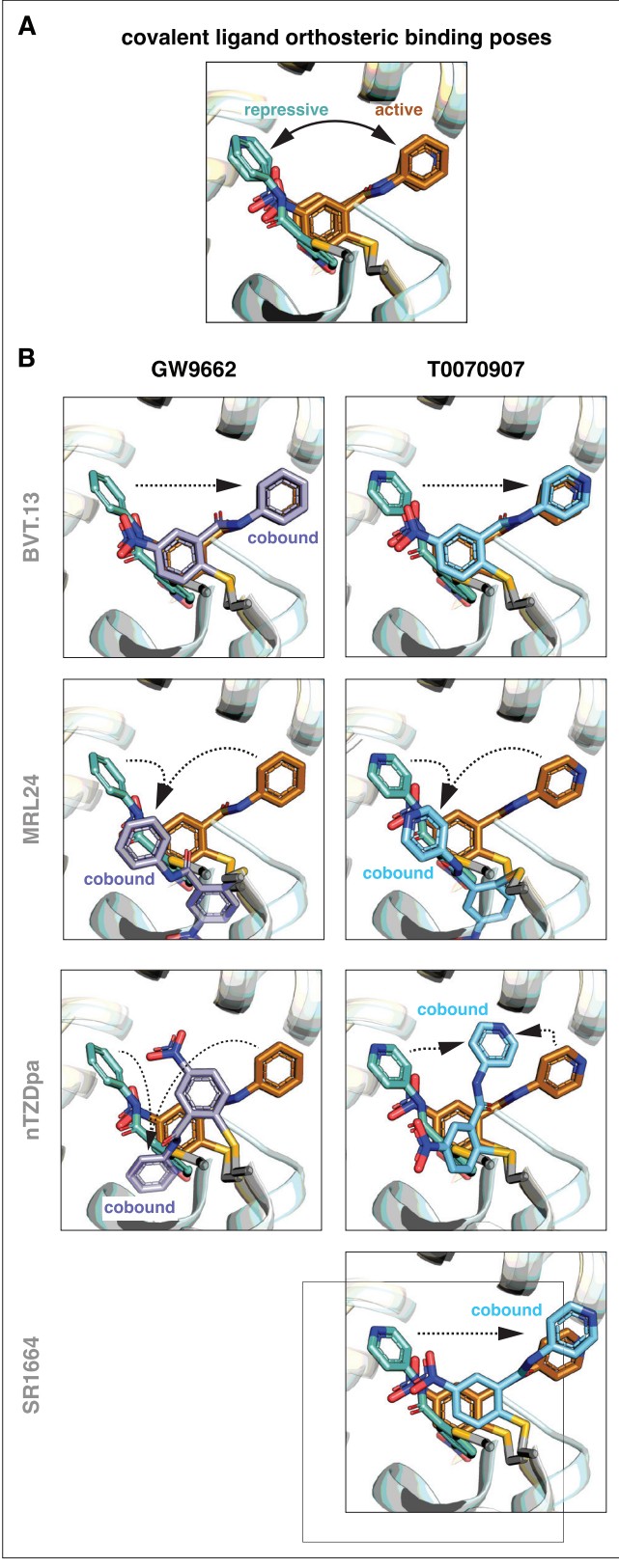

**Figure 6.** Covalent inhibitors adopt different binding modes to accommodate a cobound non-covalent synthetic ligand. (**A**) Structural overlay showing the orthosteric binding modes of GW9662 and T0070907 in crystal structures of PPARγ LBD in active and repressive conformations. (**B**) Comparison of the covalent binding modes when singly bound (orthosteric) and cobound to a non-covalent synthetic ligand. Black arrows indicate the conformational

*Figure 6 continued on next page*

*Figure 6 continued*

differences between the orthosteric binding modes vs. cobinding modes. PDB codes for crystal structures used in the overlays are listed in the Materials and methods section.

## Discussion

Crystal structures of PPARγ bound to covalent and non-covalent synthetic ligands have shown overlapping ligand binding modes, indicating the covalent ligands should block binding of the non-covalent ligands. We and others have posited that cobinding of a non-covalent synthetic ligand to the PPARγ LBD prebound to a covalent ligand (GW9662 or T0070907) would reveal an alternate site binding mode, different from the crystallized binding modes. To our surprise, the non-covalent ligand can bind via its native orthosteric binding mode by pushing the covalent ligand aside, or slightly adapt its binding mode within the orthosteric pocket. Our structural analysis of non-covalent ligand cobinding with a covalent ligand reveals several important observations and conclusions.

Each of the non-covalent synthetic ligands utilizes unique mechanisms to cobind with the covalent ligands, as inferred from our crystal structures in which pre-formed crystals of the PPARγ LBD were soaked with non-covalent synthetic ligands BVT.13 cobinds to a region of the orthosteric pocket that only slightly overlaps with the orthosteric covalent ligand binding pose, adopting a cobinding pose that is similar to when soaked into apo-PPARγ LBD crystals. MRL24 and nTZDpa cobinding pushes the covalent orthosteric ligand into a different conformation, allowing these ligands to adopt their orthosteric binding modes in the cobound state. This finding was surprising for MRL24 in particular, as its orthosteric binding mode overlaps considerably with the orthosteric covalent ligand binding mode. In contrast, SR1664 cobinding occurs to the so-called alternate site, located at the entrance to the orthosteric pocket.

In our previous study, we observed synthetic and natural/endogenous ligand co-binding via co-crystallography where preformed crystals of PPARγ LBD bound to unsaturated fatty acids (UFAs) were soaked with a synthetic ligand, which pushed the bound UFA to an alternate site within the orthosteric ligand-binding pocket (*Shang et al., 2018*). In the scenario of synthetic ligand cobinding with a covalent inhibitor, it is possible that soaking a covalent inhibitor into preformed crystals where the PPARγ LBD is already bound to a non-covalent ligand may prove to be difficult. The covalent inhibitor would need to flow through solvent channels within the crystal lattice, which may not be a problem. However, upon reaching the entrance surface to the orthosteric ligand-binding pocket, it may be difficult for the covalent inhibitor to gain access to the region of the orthosteric pocket required for covalent modification as the larger non-covalent ligand could block access. This potential order of addition problem may not be a problem for studies in solution or in cells, where the non-covalent ligand can more freely exchange in and out of the orthosteric pocket and over time the covalent reaction would reach full occupancy.

Our data provide support to structural model where in the absence of ligand, the PPARγ LBD exchanges between a transcriptionally repressive conformation where helix 12 is solvent occluded within the orthosteric pocket and a solvent exposed active conformation (*Shang et al., 2020*). T0070907 binding slows the rate of exchange between these two conformations, stabilizing a long-lived active and repressive state (*Brust et al., 2018*). GW9662-bound PPARγ LBD also samples active and repressive states, as observed by PRE NMR (*Shang et al., 2020*), although the active conformation is more abundantly populated (*Chrisman et al., 2018*). Agonist binding occurs via a two-step mechanism involving a fast encounter complex binding step at the entrance to the orthosteric pocket followed by a slow conformational change where the ligand translocates into the orthosteric pocket (*Shang and Kojetin, 2021*). Our NMR data here show that non-covalent ligand cobinding to T0070907-bound PPARγ LBD selects for an active conformation, as the repressive conformation NMR peak disappears. This indicates non-covalent ligand cobinding prevents the LBD from adopting a transcriptionally repressive helix 12 conformation within orthosteric pocket, explaining why non-covalent ligand cobinding of MRL24 and other PPARγ agonists with a covalent ligand activates PPARγ transcription (*Brust et al., 2017*; *Hughes et al., 2016*; *Hughes et al., 2014*). Another way to test this structural model could be through the use of covalent PPARγ inverse agonist analogs with graded activity (*MacTavish et al., 2024*), where one might posit that covalent inverse agonist analogs that

shift the LBD conformational ensemble towards a fully repressive LBD conformation may better inhibit synthetic ligand cobinding.

Our findings may have profound implications as these GW9662 and T0070907 have been used in many published studies as covalent inhibitors to block ligand binding to PPARγ to test for binding and functional specificity in cells. As of May 2024, more than 1900 citations referring to 'GW9662 or T0070907' are reported in PubMed. Nearly 1000 of these publications were published in 2015 or later, after the original report that GW9662 and T0070907 are not effective inhibitors of ligand binding (*Hughes et al., 2014*), yet many in the field continue to use these compounds as covalent inhibitors. Our findings strongly suggest GW9662 and T0070907 should not be used as antagonists to block binding of other synthetic PPARγ ligands. On the other hand, GW9662 and T0070907 display unique pharmacological properties as a transcriptionally neutral antagonist and repressive inverse agonist, respectively (*Brust et al., 2018*), and thus it would be appropriate to use these compounds as pharmacological ligands. It may be possible to use the crystal structures we obtained to guide structure-informed design of covalent inhibitors that would physically block cobinding of a synthetic ligand. This could be the potential mechanism of a newer generation covalent antagonist inhibitor we developed, SR16832, that more completely inhibit alternate site ligand binding of an analog of MRL20, rosiglitazone and the UFA docosahexaenoic acid (DHA) (*Brust et al., 2017*) and thus may be a better choice for the field to use as a covalent ligand inhibitor of PPARγ.

# Materials and methods

**Key resources table**

| Reagent type (species) or resource | Designation | Source or reference | Identifiers | Additional information |
|---|---|---|---|---|
| Gene | PPARG (*Homo sapiens*) | UniPro | P37231 | Protein sequence |
| Strain, strain background (*Escherichia coli*) | BL21(DE3) | Sigma-Aldrich | CMC0016 | Electrocompetent cells |
| Chemical compound | T0070907 | Cayman Chemical | 10026 | |
| Chemical compound | GW9662 | Cayman Chemical | 70785 | |
| Chemical compound | MRL-24 | MecChem Express | HY-122235 | |
| Chemical compound | nTZDpa | Tocris Bioscience | 2150 | |
| Chemical compound | SR1664 | Cayman Chemical | 11086 | |
| Chemical compound | BVT-13 | Sigma Aldrich | B4438 | |
| Recombinant DNA reagent | PPARγ LBD | *Hughes et al., 2012* | Bacterial expression plasmid | |
| Antibody | LanthaScreen Elite Tb-anti-His antibody | Thermo Fisher | #PV5895 | |
| Peptide, recombinant protein | TRAP220/MED1 | LifeTein | synthesized | residues 638–656 (NTKNHPMLMNLLKDNPAQD) synthesized with or without a N-terminal FITC label with a six-carbon linker (Ahx) and an amidated C-terminus for stability |
| Peptide, recombinant protein | NCoR1 | LifeTein | synthesized | residues 2256–2,278 (DPASNLGLEDIIRKALMGSFDDK) synthesized with or without a N-terminal FITC label with a six-carbon linker (Ahx) and an amidated C-terminus for stability |
| Software | NMRFx | *Norris et al., 2016* | Version 11.4 .x | |
| Software | Topspin | Bruker | Version 3 .x | |
| Software | Prism | GraphPad | Version 10 | |
| Software | CCP4 | *Agirre et al., 2023* | | |

*Continued on next page*

*Continued*

| Reagent type (species) or resource | Designation | Source or reference | Identifiers | Additional information |
|---|---|---|---|---|
| Software | Phaser | *McCoy et al., 2007* | | |
| Software | Phenix | *Adams et al., 2010* | | |
| Software | COOT | *Emsley and Cowtan, 2004* | | |
| Software | PyMOL | Schrödinger | Version 3 | |
| Software | jFATCAT | RCSB | | |

## Materials and reagents

All compounds used in this study—GW9662, T0070907, BVT-13, nTZDpa, MRL24, and SR1664—were obtained from commercial sources including Cayman Chemicals, Tocris Bioscience, and Sigma-Aldrich with purity >95%. Peptides of LXXLL-containing motifs from TRAP220/MED1 (residues 638–656; NTKNHPMLMNLLKDNPAQD) and NCoR1 (2256–2278; DPASNLGLEDIIRKALMGSFDDK) with or without a N-terminal FITC label with a six-carbon linker (Ahx) and an amidated C-terminus for stability were synthesized by LifeTein.

## Protein expression and purification

Human PPARγ LBD (residues 203–477, isoform 1 numbering) was expressed in *Escherichia coli* BL21(DE3) cells using autoinduction ZY media or M9 minimal media supplemented with NMR isotopes ($^{15}NH_4Cl$) as a Tobacco Etch Virus (TEV)-cleavable N-terminal His-tagged fusion protein using a pET46 Ek/LIC vector (Novagen) and purified using Ni-NTA affinity chromatography and gel filtration chromatography. The purified proteins were concentrated to 10 mg/mL in a buffer consisting of 20 mM potassium phosphate (pH 7.4), 50 mM potassium chloride, 5 mM tris(2-carboxyethyl)phosphine (TCEP), and 0.5 mM ethylenediaminetetraacetic acid (EDTA). Purified protein was verified by SDS-PAGE as >95% pure. For studies using a covalent orthosteric antagonist, PPARγ LBD protein was incubated with at least a~1.05 x excess of GW9662 or T0070907 at 4 °C for 24 hr to ensure covalent modification to residue C285, then buffer exchanged the sample to remove excess covalent antagonist and DMSO. Complete attachment of the covalent antagonist occurs within 30–60 min, as detected using an LTQ XL linear Ion trap mass spectrometer with an electrospray ionization source (Thermo Fisher Scientific).

## Crystallization and structure determination

For T0070907- and GW9662-bound PPARγ LBD complexes, protein was incubated at a 1:3 protein/ligand molar ratio in PBS overnight. Proteins were buffer exchanged to remove DMSO and concentrated to 10 mg/mL. All crystals were obtained after 3–8 days at 22 °C by sitting-drop vapor diffusion against 50 µL of well solution using 96-well format crystallization plates. The crystallization drops contained 1 µL of protein complex sample mixed with apo crystal seeds prepared by PTFE seed bead (Hampton research) and 1 µL of reservoir solution containing 0.1 M MOPS (pH 7.6) and 0.8 M sodium citrate for T0070907 or GW9662-PPARγ LBD complexes; 0.1 M MES (pH 6.5), 0.2 M ammonium sulfate. The non-covalent ligands (BVT.13, MRL24, nTZDpa, SR1664) were soaked into T0070907 or GW9662-PPARγ LBD complex crystals by adding 1.5 µL of compound at a concentration of 2 mM suspended in reservoir solution containing 5% DMSO for 5 days. Data were processed, integrated, and scaled with the programs Mosflm and Scala in CCP4 (*Agirre et al., 2023*). The structure was solved by molecular replacement using the program Phaser (*McCoy et al., 2007*) implemented in the PHENIX package (*Adams et al., 2010*) and used previously published PPARγ LBD structure (PDB code: 1PRG/6ONI; *Nolte et al., 1998*; *Shang et al., 2020*) as the search model. The structure was refined using PHENIX with several cycles of interactive model rebuilding in COOT (*Emsley and Cowtan, 2004*).

## NMR spectroscopy

2D [$^1$H,$^{15}$N]-TROSY HSQC NMR data of 200 µM $^{15}$N-labeled PPARγ LBD, pre-incubated with a 2 x molar excess of covalent ligand overnight at 4 °C, were acquired at 298 K on a Bruker 700 MHz NMR

instrument equipped with a QCI cryoprobe in NMR buffer (50 mM potassium phosphate, 20 mM potassium chloride, 1 mM TCEP, pH 7.4, 10% $D_2O$). Data were processed and analyzed using Topspin 3.0 (Bruker Biospin) and NMRViewJ (OneMoon Scientific, Inc; *Johnson, 2018*), respectively. NMR chemical shift assignments previously transferred from rosiglitazone-bound PPARγ LBD (*Hughes et al., 2012*) to T0070907- and GW9662-bound states (*Brust et al., 2018*; *Shang et al., 2020*) were used in this study for well-resolved residues with conversed NMR peak positions to the previous ligand-bound forms using the minimum chemical shift perturbation procedure (*Williamson, 2013*).

## Time-resolved fluorescence resonance energy transfer (TR-FRET) assay

The time-resolved fluorescence resonance energy transfer (TR-FRET) assays were performed in black 384-well plates (Greiner) with 23 μL final well volume containing 4 nM His$_6$-PPARγ LBD with or without covalently modification by GW9662 or T0070907, 1 nM LanthaScreen Elite Tb-anti-His Antibody (Thermo Fisher), and 400 TRAP220 or NCoR peptide in TR-FRET buffer (20 mM $KPO_4$ pH 7.4, 50 mM KCl, 5 mM TCEP, 0.005% Tween 20). Compound stocks were prepared via serial dilution in DMSO, added to wells in triplicate, and plates were read using BioTek Synergy Neo multimode plate reader after incubating at 25 °C for 1 hr. The Tb donor was excited at 340 nm, its emission was measured at 495 nm, and the acceptor FITC emission was measured at 520 nm. Data were plotted using GraphPad Prism as TR-FRET ratio 520 nm/495 nm vs. ligand concentration and fit to sigmoidal dose-response equation — or biphasic or bell shaped dose response equations when appropriate, determined by comparison of fits to both equations and F test where the simpler model is selected if the p value is less than 0.05.

## Structural comparisons to published crystal structures

Structural overlays were compared to crystals structures of PPARγ LBD bound to GW9662 (PDB 3B0R), T0070907 (PDB 6C1I; *Brust et al., 2018*), GW9662 and NCoR1 peptide (PDB 8FHE; *MacTavish et al., 2024*), T0070907 and NCoR1 peptide (6ONI) *Shang et al., 2020*, BVT.13 (PDB 2Q6S) *Bruning et al., 2007*, MRL24 (PDB 2Q5P; *Bruning et al., 2007*), nTZDpa (PDB 2Q5S; *Bruning et al., 2007*), SR1664 and NCOA1 peptide (PDB 5DWL; *Bae et al., 2016*), and SR1664 (PDB 4R2U; *Marciano et al., 2015*). Pairwise structural alignment and rmsd calculations of the cobound structures to the transcriptionally active (PDB 6ONJ) PPARγ LBD conformation was performed via the RCSB webserver (https://www.rcsb.org/alignment/) using the jFATCAT rigid structural alignment algorithm.

## Acknowledgements

This work was supported in part by the National Institutes of Health (NIH) grant R01DK124870 from the National Institute of Diabetes and Digestive and Kidney Diseases (NIDDK). Use of the Stanford Synchrotron Radiation Lightsource, SLAC National Accelerator Laboratory, is supported by the U.S. Department of Energy, Office of Science, Office of Basic Energy Sciences under Contract No. DE-AC02-76SF00515. The SSRL Structural Molecular Biology Program is supported by the DOE Office of Biological and Environmental Research and by the NIH National Institute of General Medical Sciences (NIGMS) grant P30GM133894. The contents of this publication are solely the responsibility of the authors and do not necessarily represent the official views of NIDDK, NIGMS, NIH, or DOE.

## Additional information

### Funding

| Funder | Grant reference number | Author |
| --- | --- | --- |
| National Institute of Diabetes and Digestive and Kidney Diseases | R01DK124870 | Douglas J Kojetin |
| National Natural Science Foundation of China | 82170473 | Jinsai Shang |

| Funder | Grant reference number | Author |
|---|---|---|

The funders had no role in study design, data collection and interpretation, or the decision to submit the work for publication.

## Author contributions

Jinsai Shang, Conceptualization, Formal analysis, Investigation, Visualization, Methodology, Writing – original draft, Writing – review and editing; Douglas J Kojetin, Conceptualization, Formal analysis, Supervision, Funding acquisition, Visualization, Methodology, Writing – original draft, Writing – review and editing

## Author ORCIDs

Jinsai Shang ⓘ https://orcid.org/0000-0001-8164-1544
Douglas J Kojetin ⓘ https://orcid.org/0000-0001-8058-6168

Reviewer #1 (Public review): https://doi.org/10.7554/eLife.99782.3.sa1
Reviewer #2 (Public review): https://doi.org/10.7554/eLife.99782.3.sa2
Author response https://doi.org/10.7554/eLife.99782.3.sa3

# Additional files

## Supplementary files

MDAR checklist

## Data availability

Diffraction data and crystal structures have been deposited in the PDB under accession codes 8ZFN, 8ZFO, 8ZFP, 8ZFQ, 8ZFR, 8ZFS, and 8ZFT. Other data generated and analyzed during the current study are in *Figure 3—source data 1*.

The following datasets were generated:

| Author(s) | Year | Dataset title | Dataset URL | Database and Identifier |
|---|---|---|---|---|
| Shang J, Kojetin DJ | 2024 | Crystal Structure of Human PPARgamma Ligand Binding Domain in Complex with GW9662 and BVT.13 | https://www.wwpdb.org/pdb?id=pdb_00008zfn | RCSB Protein Data Bank, 8ZFN |
| Shang J, Kojetin DJ | 2024 | Crystal Structure of Human PPARgamma Ligand Binding Domain in Complex with GW9662 and nTZDpa | https://www.wwpdb.org/pdb?id=pdb_00008zfo | RCSB Protein Data Bank, 8ZFO |
| Shang J, Kojetin DJ | 2024 | Crystal Structure of Human PPARgamma Ligand Binding Domain in Complex with GW9662 and MRL24 | https://www.wwpdb.org/pdb?id=pdb_00008zfp | RCSB Protein Data Bank, 8ZFP |
| Shang J, Kojetin DJ | 2024 | Crystal Structure of Human PPARgamma Ligand Binding Domain in Complex with T0070907 and BVT | https://www.wwpdb.org/pdb?id=pdb_00008zfq | RCSB Protein Data Bank, 8ZFQ |
| Shang J, Kojetin DJ | 2024 | Crystal Structure of Human PPARgamma Ligand Binding Domain in Complex with T0070907 and nTZDpa | https://www.wwpdb.org/pdb?id=pdb_00008zfr | RCSB Protein Data Bank, 8ZFR |

*Continued*

| Author(s) | Year | Dataset title | Dataset URL | Database and Identifier |
|---|---|---|---|---|
| Shang J, Kojetin DJ | 2024 | Crystal Structure of Human PPARgamma Ligand Binding Domain in Complex with T0070907 and MRL24 | https://www.wwpdb.org/pdb?id=pdb_00008zfs | RCSB Protein Data Bank, 8ZFS |
| Shang J, Kojetin DJ | 2024 | Crystal Structure of Human PPARgamma Ligand Binding Domain in Complex with T0070907 and SR1664 | https://www.wwpdb.org/pdb?id=pdb_00008zft | RCSB Protein Data Bank, 8ZFT |

The following previously published datasets were used:

| Author(s) | Year | Dataset title | Dataset URL | Database and Identifier |
|---|---|---|---|---|
| Tomioka D, Hashimoto H, Sato M, Shimizu T | 2011 | Human PPAR gamma ligand binding dmain complexed with GW9662 in a covalent bonded form | https://www.wwpdb.org/pdb?id=pdb_00003b0r | RCSB Protein Data Bank, 3B0R |
| Shang J, Fuhrmann J, Brust R, Kojetin DJ | 2018 | Crystal Structure of Human PPARgamma Ligand Binding Domain in Complex with T0070907 | https://www.wwpdb.org/pdb?id=pdb_00006c1i | RCSB Protein Data Bank, 6C1I |
| Shang J, Kojetin DJ | 2022 | Crystal structure of PPARgamma ligand-binding domain in complex with N-CoR peptide and GW9662 | https://www.wwpdb.org/pdb?id=pdb_00008fhe | RCSB Protein Data Bank, 8FHE |
| Shang J, Kojetin DJ | 2019 | Crystal structure of PPARgamma ligand binding domain in complex with N-CoR peptide and inverse agonist T0070907 | https://www.wwpdb.org/pdb?id=pdb_00006oni | RCSB Protein Data Bank, 6ONI |
| Bruning JB, Nettles KW | 2007 | 2.4 angstrom crystal structure of PPAR gamma complexed to BVT.13 without co-activator peptides | https://www.wwpdb.org/pdb?id=pdb_00002q6s | RCSB Data Bank, 2Q6S |
| Bruning JB, Nettles KW | 2007 | Crystal Structure of PPARgamma bound to partial agonist MRL24 | https://www.wwpdb.org/pdb?id=pdb_00002q5p | RCSB Protein Data Bank, 2Q5P |
| Bruning JB, Nettles KW | 2007 | Crystal Structure of PPARgamma bound to partial agonist nTZDpa | https://www.wwpdb.org/pdb?id=pdb_00002q5s | RCSB Protein Data Bank, 2Q5S |
| Jang JY | 2015 | Human PPARgamma ligand binding dmain in complex with SR1664 | https://www.wwpdb.org/pdb?id=pdb_00005dwl | RCSB Protein Data Bank, 5DWL |
| Marciano DP, Kamenecka T, Griffin PR, Bruning JB | 2014 | Crystal Structure of PPARgamma in complex with SR1664 | https://www.wwpdb.org/pdb?id=pdb_00004r2u | RCSB Protein Data Bank, 4R2U |
| Shang J, Kojetin DJ | 2019 | Crystal structure of PPARgamma ligand binding domain in complex with TRAP220 peptide and agonist rosiglitazone | https://www.wwpdb.org/pdb?id=pdb_00006onj | RCSB Protein Data Bank, 6ONJ |

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
