## [Editor Report · eLife Assessment]

This **landmark** study elucidates the intricate structural mechanisms by which both covalent and non-covalent synthetic ligands can co-occupy the binding pocket of the nuclear receptor transcription factor PPARγ. Through a **compelling** integration of structural, biochemical, and biophysical evidence, the authors challenge the reliability of two commonly used covalent inhibitors. These findings have far-reaching implications for the broader field of nuclear receptor research. This work will be of high interest to structural biologists and biochemists exploring ligand interactions within the nuclear receptor superfamily.

---

## [Referee Report · Reviewer #1 (Public review)]

Summary:

PPARgamma is a nuclear receptor that binds to orthosteric ligands to coordinate transcriptional programs that are critical for adipocyte biogenesis and insulin sensitivity. Consequently, it is a critical therapeutic target for many diseases but especially diabetes. The malleable nature and promiscuity of the PPARgamma orthosteric ligand binding pocket has confounded the development of improved therapeutic modulators. Covalent inhibitors have been developed but they show unanticipated mechanisms of action depending on which orthosteric ligands are present. In this work, Shang and Kojetin present a compelling and comprehensive structural, biochemical, and biophysical analysis that shows how covalent and noncovalent ligands can co-occupy the PPARgamma ligand binding pocket to elicit distinctive preferences of coactivator and corepressor proteins. Importantly, this work shows how the covalent inhibitors GW9662 and T0070907 may be unreliable tools as pan-PPARgamma inhibitors despite their wide-spread use.

Strengths:

- Highly detailed structure and functional analyses provide a comprehensive structure-based hypothesis for the relationship between PPARgamma ligand binding domain co-occupancy and allosteric mechanisms of action.

- Multiple orthogonal approaches are used to provide high resolution information on ligand binding poses and protein dynamics.

- The large number of x-ray crystal structures solved for this manuscript should be applauded along with their rigorous validation and interpretation.

---

## [Referee Report · Reviewer #2 (Public review)]

Summary:

The flexibility of the ligand binding domain (LBD) of NRs allows various modes of ligand binding leading to various cellular outcomes. In the case of PPARγ, it's known that two ligands can cobind to the receptor. However, whether a covalent inhibitor functions by blocking the binding of a non-covalent ligand, or cobind in a manner that weakens the binding of a non-covalent ligand remains unclear. In this study, the authors first used TR-FRET and NMR to demonstrate that covalent inhibitors (such as GW9662 and T0070907) weaken but do not prevent non-covalent synthetic ligands from binding, likely via an allosteric mechanism. The AF-2 helix can exchange between active and repressive conformations, and covalent inhibitors shift the conformation toward a transcriptionally repressive one to reduce orthosteric binding of the non-covalent ligands. By co-crystal studies, the authors further reveal the structural details of various non-covalent ligand binding mechanisms in a ligand specific manner (e.g., an alternate binding site, or a new orthosteric binding mode by alerting covalent ligand binding pose).

Strengths:

The biochemical and biophysical evidence as presented is strong and convincing.

Additional comments:

The co-crystal studies were performed by soaking a non-covalent ligand to LBD pre-crystalized with a covalent inhibitor. Since the covalent inhibitors would shift the LBD toward transcriptionally repressive conformation which reduces orthosteric binding of non-covalent ligands, one might ask if the sequence was reversed (i.e., soaking a covalent inhibitor to LBD pre-crystalized with a non-covalent ligand), would similar conclusion be drawn? The authors have reasonably speculated that it might be difficult to soak a covalent inhibitor into preformed crystals where the PPARγ LBD is already bound to a non-covalent ligand, because the larger non-covalent ligand could block the covalent inhibitor to gain access to the region of the orthosteric pocket required for covalent modification.

---

## [Author Response]

The following is the authors’ response to the original reviews.

**Public Reviews:**

**Reviewer #1 (Public Review):**
Summary:PPARgamma is a nuclear receptor that binds to orthosteric ligands to coordinate transcriptional programs that are critical for adipocyte biogenesis and insulin sensitivity. Consequently, it is a critical therapeutic target for many diseases, but especially diabetes. The malleable nature and promiscuity of the PPARgamma orthosteric ligand binding pocket have confounded the development of improved therapeutic modulators. Covalent inhibitors have been developed but they show unanticipated mechanisms of action depending on which orthosteric ligands are present. In this work, Shang and Kojetin present a compelling and comprehensive structural, biochemical, and biophysical analysis that shows how covalent and noncovalent ligands can co-occupy the PPARgamma ligand binding pocket to elicit distinctive preferences of coactivator and corepressor proteins. Importantly, this work shows how the covalent inhibitors GW9662 and T0070907 may be unreliable tools as pan-PPARgamma inhibitors despite their widespread use.Strengths:- Highly detailed structure and functional analyses provide a comprehensive structure-based hypothesis for the relationship between PPARgamma ligand binding domain co-occupancy and allosteric mechanisms of action. - Multiple orthogonal approaches are used to provide high-resolution information on ligand binding poses and protein dynamics.- The large number of x-ray crystal structures solved for this manuscript should be applauded along with their rigorous validation and interpretation.Weaknesses- Inclusion of statistical analysis is missing in several places in the text. - Functional analysis beyond coregulator binding is needed.

We added additional statistical analyses as recommended (Source Data 1, a Microsoft Excel spreadsheet).

Related to functional analysis, we cite and studies from our previous publication (Hughes et al. Nature Communications 2014 5:3571) where we demonstrated that the covalent inhibitor ligands (GW9662 and T0070907) do not block the activity of other ligands using a PPARγ transcriptional reporter assay and gene expression analysis in 3T3-L1 preadipocytes. Our study here expands on this finding and other published studies showing the structural mechanism for the lack of blocking activity by the covalent inhibitors.

**Reviewer #2 (Public Review):**
Summary:The flexibility of the ligand binding domain (LBD) of NRs allows various modes of ligand binding leading to various cellular outcomes. In the case of PPARγ, it's known that two ligands can co-bind to the receptor. However, whether a covalent inhibitor functions by blocking the binding of a non-covalent ligand, or co-bind in a manner that weakens the binding of a non-covalent ligand remains unclear. In this study, the authors first used TR-FRET and NMR to demonstrate that covalent inhibitors (such as GW9662 and T0070907) weaken but do not prevent non-covalent synthetic ligands from binding, likely via an allosteric mechanism. The AF-2 helix can exchange between active and repressive conformations, and covalent inhibitors shift the conformation toward a transcriptionally repressive one to reduce the orthosteric binding of the non-covalent ligands. By co-crystal studies, the authors further reveal the structural details of various non-covalent ligand binding mechanisms in a ligand-specific manner (e.g., an alternate binding site, or a new orthosteric binding mode by alerting covalent ligand binding pose).Strengths:The biochemical and biophysical evidence presented is strong and convincing.Weaknesses:However, the co-crystal studies were performed by soaking non-covalent ligands to LBD pre-crystalized with a covalent inhibitor. Since the covalent inhibitors would shift the LBD toward transcriptionally repressive conformation which reduces orthosteric binding of non-covalent ligands, if the sequence was reversed (i.e., soaking a covalent inhibitor to LBD pre-crystalized with a non-covalent ligand), would a similar conclusion be drawn? Additional discussion will broaden the implications of the conclusion.

This is an interesting point, which we now expand upon in a new (third) paragraph of the discussion in our revised manuscript:

“In our previous study, we observed synthetic and natural/endogenous ligand co-binding via co-crystallography where preformed crystals of PPARγ LBD bound to unsaturated fatty acids (UFAs) were soaked with a synthetic ligand, which pushed the bound UFA to an alternate site within the orthosteric ligand-binding pocket 8. In the scenario of synthetic ligand cobinding with a covalent inhibitor, it is possible that soaking a covalent inhibitor into preformed crystals where the PPARγ LBD is already bound to a non-covalent ligand may prove to be difficult. The covalent inhibitor would need to flow through solvent channels within the crystal lattice, which may not be a problem. However, upon reaching the entrance surface to the orthosteric ligand-binding pocket, it may be difficult for the covalent inhibitor to gain access to the region of the orthosteric pocket required for covalent modification as the larger non-covalent ligand could block access. This potential order of addition problem may not be a problem for studies in solution or in cells, where the non-covalent ligand can more freely exchange in and out of the orthosteric pocket and over time the covalent reaction would reach full occupancy.”

**Recommendations for the authors:**

**Reviewer #1 (Recommendations For The Authors):**
- IC50 or EC50 values are not reported for the coregulator interaction assays, R2 for fit should also be reported where Ki and IC50s are disclosed.

We now report fitting statistics and IC50/EC50 values when possible in Figure 2B and Source Data 1 along with R2 values for the fit. We note that some data do not show complete or robust enough binding curves to faithfully fit to a dose response equation.

- Reporter gene or qPCR should be performed for the combinations of covalent and noncovalent ligands to show how these molecules impact transcriptional activities rather than just coregulator binding profiles.

We previously performed PPARγ transcriptional reporter assay and gene expression analysis in 3T3-L1 preadipocytes to demonstrate that cotreatment of a covalent inhibitor (GW9662 or T0070907) with a non-covalent ligand does not block activity of the non-covalent ligand and showed cobinding-induced activation relative to DMSO control (Hughes et al., 2024 Nature Communications). We did not specifically mention this in our original manuscript, but we now call this out in the first paragraph of the results section.

- Inclusion of a structure figure to show the different helix 12 orientations should be included in the introduction. Likewise, how the overall structure of the LBD changes as a result of the cobinding in the discussion or a summary model would be helpful.

Our revised manuscript includes a structure figure called out in the introduction describing the active and repressive helix 12 PPARγ LBD conformations (new Figure 1). There are no major changes to the overall structure of the LBD compared to the active conformation that crystallized, so we did not include a summary model figure but we do refer readers to our previous paper (Shang and Kojetin, Structure 2021 29(9):940-950) in the penultimate paragraph of the discussion. We also added the following sentence to the crystallography results section related to the overall LBD changes:

“The structures show high structural similarity to the transcriptionally active LBD conformation with rmsd values ranging from 0.77–1.03Å (Supplementary Table S2)”

A typo in paragraph 3 of the discussion says "long-live" when it should probably say "long-lived."

We corrected this typo.

**Reviewer #2 (Recommendations For The Authors):**
It's interesting that ligand-specific binding mode of non-covalent ligands was observed. Would modifications of the chemical structure of a covalent inhibitor alter the allosteric binding behavior of non-covalent ligands in a predictive manner? If so, how can such SAR be used to guide the design of covalent inhibitors to more broadly and effectively inhibit agonists of various chemical structures? Discussion on this topic could be valuable.

This is an interesting point, which we now discuss in the penultimate and last paragraphs of the discussion:

“Another way to test this structural model could be through the use of covalent PPARγ inverse agonist analogs with graded activity 23, where one might posit that covalent inverse agonist analogs that shift the LBD conformational ensemble towards a fully repressive LBD conformation may better inhibit synthetic ligand cobinding.”

“It may be possible to use the crystal structures we obtained to guide structure-informed design of covalent inhibitors that would physically block cobinding of a synthetic ligand. This could be the potential mechanism of a newer generation covalent antagonist inhibitor we developed, SR16832, that more completely inhibit alternate site ligand binding of an analog of MRL20, rosiglitazone and the UFA docosahexaenoic acid (DHA)

21 and thus may be a better choice for the field to use as a covalent ligand inhibitor of PPARγ.”